# Impact of Methionine Synthase Reductase Polymorphisms in Chronic Myeloid Leukemia Patients

**DOI:** 10.3390/genes13101729

**Published:** 2022-09-26

**Authors:** Abozer Y. Elderdery, Entesar M. Tebein, Fawaz O. Alenazy, Ahmed M. E. Elkhalifa, Manar G. Shalabi, Anass M. Abbas, Hassan H. Alhassan, Chand B. Davuljigari, Jeremy Mills

**Affiliations:** 1Department of Clinical Laboratory Sciences, College of Applied Medical Sciences, Jouf University, Sakaka 72388, Saudi Arabia; 2Health Sciences Research Unit, Jouf University, Sakaka 72388, Saudi Arabia; 3College of Applied Medical Sciences, Shaqra University, Shaqra 11961, Saudi Arabia; 4Department of Public Health, College of Health Sciences, Saudi Electronic University, Riyadh 11673, Saudi Arabia; 5Department of Haematology, Faculty of Medical Laboratory Sciences, University of El Imam El Mahdi, Kosti 11588, Sudan; 6Department of Zoology, College of Sciences, Sri Venkateswara University, Tirupati 517502, Andhra Pradesh, India; 7School of Pharmacy and Biomedical Sciences, University of Portsmouth, Portsmouth PO1 2UP, UK

**Keywords:** CML, *MTRR*, folate and genetic polymorphism, Sudanese population

## Abstract

**Highlights:**

**What are the main findings?**
Our objective was to determine the relationship between the methionine synthase reductase polymorphisms (*MTRR* and *MTR*) and the risk of developing chronic myeloid leukemia (CML) in Sudanese Patients.

**What is the implication of the main finding?**
Here, we report that the heterozygous and homozygous mutant genotypes of *MTRR* polymorphisms were associated with a decreased risk of developing CML in the Sudanese population asa protective factor.Our findings will help to increase the understanding of *MTRR* A66G polymorphism and its association with CML risk in the Sudanese population.

**Abstract:**

**Introduction：** Metabolism methionine and of folate play a vital function in cellular methylation reactions, DNA synthesis and epigenetic process.However, polymorphisms of methionine have received much attention in recent medical genetics research. **Objectives**: To ascertain whether the common polymorphisms of the *MTRR* (Methionine Synthase Reductase) A66G gene could play a role in affecting susceptibility to Chronic Myeloid Leukemia (CML) in Sudanese individuals. **Methods:** In a case-controlled study, we extracted and analyzed DNA from 200 CML patients and 100 healthy control subjects by the PCR-RFLP method. **Results:** We found no significant difference in age orgender between the patient group and controls. The *MTRR* A66G genotypes were distributed based on the Hardy-Weinberg equilibrium (*p* > 0.05). The variation of *MTRR* A66G was less significantly frequent in cases with CML (68.35%) than in controls (87%) (OR = 0.146, 95% CI = 0.162–0.662, *p* < 0.002). Additionally, AG and GG genotypes and G allele were reducing the CML risk (Odds ratio [OR] = 0.365; 95% CI [0.179–0.746]; *p* = 0.006; OR = 0.292; 95% CI [0.145–0.590]; *p* = 0.001 and OR = 0.146; 95% CI [0.162–0.662]; *p* = 0.002 and OR = 2.0; 95% CI [1.3853–2.817]; respectively, (*p* = 0.000)). **Conclusions:** Our data demonstrated that heterozygous and homozygous mutant genotypes of *MTRR* polymorphisms were associated with decreased risk of developing CML in the Sudanese population.

## 1. Introduction

Chronic myeloid leukemia (CML) is a neoplasm disorder derived from stem cells and characterized by the Philadelphia (*Ph*) chromosome, developedafter a reciprocal translation between chromosomes 9 and 22, which leads to the hybridization of the BCR (breakpoint cluster region) and ABL (Abelson oncogene locus) [1]. Additional chromosomal changes (ACAs) in CML often occur in Philadelphia chromosome (*Ph*)-positive cells and are linked with disease acceleration and treatment resistance [2]. The Ph-negative clone has a genetic abnormality and chromosomal changes that induces *t* (9;22) [3]. This rearrangement of genes encourages a conformational discrepancy in ABL protein, which makes it a hyperactive tyrosine kinase enzyme. The tyrosine kinase activity of the BCR-ABL fusion oncogene prevents apoptosis and activates the pathophysiological phases of CML [4].

The cancer death rate is increasing worldwide, including from leukemia [5]. In Sudan, CML incidence has increased considerably in recent years. It is the most common type of leukemia in age groups above 15 years; however, of all leukemia types, it is the second most common malignancy [6,7]. A recent study reported that diet and physical activities are both factors contributing to the rise of cancer [8]. Kawakita and co-workers documented an association between cancer risk and folate intake [9]. In addition, the interaction between genetic factors and environmental contact is hypothesized to be a cause of various types of cancer [10].

The reasons and mechanisms of CML are still incompletely known. However, it is well known that the vulnerability to the identical type of cancer varies between individuals even with similar environmental exposures [11]. This difference may be explained by genetic susceptibility including polymorphisms in the genes involved in the etiology of CML. Polymorphisms in the genes coding for detoxification enzymes cause variations between individuals, contributing to leukemia predisposition [12]. Genetic polymorphism is highly focused upon in recent medical genetics research [13]. Several studies are performed using genetic polymorphisms involved in folic acid metabolism in numerous forms of cancer [14,15,16]. This is because the metabolism of both methionine and folic acid plays a vital role in epigenetic processes, the synthesis of DNA, and cellular methylation reactions [17].

The metabolic pathway of folic acid is generally believed to possess a vital role in leukemia progression, as it delivers one-carbon donors for purine and pyrimidine synthesis, plus the methylation reactions and remethylation (of homocysteine) involved in DNA methylation [18]. However, many enzymes are involved in folic acid metabolism including *MTRR* (Methionine Synthase Reductase), MTHFR (Methylenetetrahydrofolate Reductase Enzyme), and MTR (Methionine Synthase) [14]. *MTRR* (Methionine Synthase Reductase) plays a vital role in folate-dependent homocysteine re-methylation and is necessary for *MTRR* activity regulation [19]. *MTRR* is one of the key regulatory enzymes participating in the folate metabolic pathway. Its polymorphism is the result of an amino acid substitution from methionine to isoleucine at codon 22, position 66, creating a G-to-A transition [20]. The *MTRR* gene is positioned on chromosome 5 at 5p15.2-p15.3 (rs1801394). The most frequent polymorphism in the *MTRR* gene is the substitution of isoleucine with methionine at position 22 (A66G; rs1801394). Gaughan et al. [21] reported that the polymorphism (MTRR A66G; rs1801394) in the *MTRR* gene is associated with decreased enzyme affinity for MS, which is closely associated with leukemia risk [21]. However, polymorphism in the *MTRR* gene has been widely investigated for the risk of leukemia. Previous studies suggest an association between *MTRR* genes and their polymorphism, and cancer risks [19,22], such as acute leukemias (lymphoblastic andmyeloid) and lung and Colorectal Cancer [20,23,24,25], but the results remain uncertain [19]. A limited number of studies investigate the influence of *MTRR* on the risk of CML predisposition [18,24]. However, inconsistent results have been reported regarding associations between this polymorphism and the risk of developing leukemia and remain controversial. To further elucidate the association of leukemia risk with MTRR A66G polymorphism, we conducted a research study on A66G polymorphism in an eligible Sudanese population.

## 2. Methodology

A case-control study was carried out at the cytogenetic lab, Radiation and Isotopes Hospital, Khartoum, Sudan, in the period of 2018/01–2018/08. All participants in the two groups signed an informed consent form. A questionnaire was used to collect data on age, gender, parity, and other characteristics. Three hundred participants were recruited, comprising 200 patients [68 females and 132 males; mean age = 45.1 (±12.3 years)] and 100 healthy individuals [49 females and 51 males; mean age = 39.2. (±13.3 years)]. All participants (cases and controls) were Sudanese, not suffering from certain diseases such as chronic myelomonocytic leukemia or other myeloproliferative disorders, and were negative for the Philadelphia chromosome (*Ph*).

CML was diagnosed based on the high WBC count (Leukocytosis) at varying degrees, as well asblood morphology (immature granulocytes, basophilia, and absolute eosinophilia) and detection of BCR-ABL by transcript analysis (all cases were positive *Ph*). The CML cases were further classified as having chronic, accelerated, or blast phases. In total, 99% of cases had the chronic phase, only one patient had the accelerated phase and no patients had the blast phase. A volume of 3 mL of venous blood from each participant was collected into EDTA containers, and the DNA was extracted using the Quinidine chloride procedure [26].

### 2.1. Genotype Analysis of MTRR A66G

The mutation was the *MTRR* A66G detected, using the RFLP (Restriction Fragment Length Polymorphism Method). PCR amplification was performed using specific forward and reverse published primers [27]. Next, 5 μL of the reaction product was electrophoresed on a 2% iNtRON agarose gel previously stained with EtBr for 45 min at 89 V and 25 m. A 50 bp biomarker ladder was added to an estimated 66-bp of amplicon and visualized on the gel documentation system previously mentioned. The amplicon (66-bp) was digested overnight at a temperature of 37 °C with the restriction enzyme Nde1 (New England BioLabs, Inc., Ipswich, MA, USA). The mixture reaction produced digested products that are visualized under electrophoresis on a 4% of iNtRON agarose gel with EtBr, in addition to a 50 bp ladder marker (iNtRON BIOTECHNOLOGY, Inc., Gyeonggi-do, Korea). The G allele produced a 66-bp band, and the A allele produced 44 and 22 bp fragments (Figure 1a,b). The sample size was calculated with a 1:2 ratio for the cases and controls and the difference in the proportions of polymorphism of *MTRR* A66G between the study groups.

### 2.2. Ethics

The study obtained ethical clearance from the ethics committee of AL-Neelain University, faculty of Medical Laboratory Science. All participants in this study signed in the consent form prior to enrollment. All protocols and procedures performed in this study were in accordance with the Declaration of Helsinki 1964 and its later amendments (revision of Edinburgh 2000). The quality of the study was examined using the Newcastle-Ottawa Scale (NOS). This scale ranges from 1 to 9. The rating system has scores ranging from zero (worst) to 9 (best), 5–6 and ≥7 stars indicate moderate and high quality, respectively.

### 2.3. Statistical Analysis

Regarding ages, the *p*-value is generated by the Kruskal-Wallis H test because age lacks normal distribution. The test for normality was performed on quantitative data, such as mean average and SD (Standard Deviation). Prior to analysis, allele frequencies were evaluated for HWE (Hardy-Weinberg Equilibrium). Groups were compared with Chi-square testing. Comparison between groups was done using ANOVA and Kruskal Wallis, the latter for non-parametric data. ORs (Odds Ratios) were given within CIs (Confidence Intervals of 95%, obtained by logistic regression. The *p*-values under 0.05 were deemed statistically significant. Statistics were performed using Statistical Package for Social Science (SPSS, Inc., Chicago, IL, USA).

## 3. Results

There were no significant variations regarding age, gender, and other characteristics between the study groups, CML patients, and controls (Table 1). The distribution of the genotype *MTRR* A66G was in accordance with the Hardy–Weinberg equilibrium (*p* > 0.05). Table 2 demonstrates the distribution of *MTRR* A66G genotypes and alleles in the study population. The *MRR* 66AA genotype frequency was higher in CML patients (31.5%) when compared to controls (13.0%). Furthermore, for controls, the frequency of the 66AG genotype (39.0%) was higher than in CML patients (34.5%) with statistical significance (*p* = 0.006) and odds ratio 0.365. So far, the individuals carrying genotype AG were protected from the risk of CML (OR = 0.365, 95%CI: 0.179–0.746, *p* = 0.006). However, the frequency of the 66GG genotype was significantly higher (OR = 0.292, 95% CI: 0.145–0.590, *p* = 0.001) among controls (48.0%) when compared to the CML group (34.0%). the GG genotype also reduces the predisposition to CML. While the G allele was higher in the healthy group (67.5%) when compared with cases (51.25%), with a statistically significant difference (*p* = 0.000), as well as G allele was reducing the CML risk (OR = 0.506, 95% CI: (0.355–0.7292), *p* = 0.000).

## 4. Discussion

Our study is the first to report that *MTRR* polymorphisms are associated with reduced risk of CML among the Sudanese, as it revealed that the calculation of OR between case and control groups for individuals with GG or AG genotype showed they were protected from CML risk with fold 3.4 and 2.7 protections, respectively. These findings are partially similar to that of a study from residents of European and Russian populations which reported a lower prevalence of the *MTRR* 66G/G genotype in acute leukemia patients than in healthy patients (*p* = 0.0015) [28]. A meta-analysis by Fang and co-workers concluded that among European individuals, especially young persons with lymphoblastic leukemia, the genotype of *MTRR*-(A66G-GG) was linked to lower leukemia risk [14]. Additionally, another study showed that *MTRR* A66G polymorphisms function as protector factors for neural tube defects (NTDs) [29]. Thus, all these findings, including ours, explain that *MTRR* polymorphism may lead to hypo-methylation which prevents DNA alteration and cell proliferation [30].

In contrast, several studies showed a significant relationship between the *MTRR* 66 A > G (rs1801394) variant and an increased risk of cancer [19,23,24,25]. Gong et al. [31] demonstrated a higher mutant G allele presence of *MTRR* A66G in a group of patients with cervical carcinoma than in normal controls; thus, confirming the association with increased risk. More significant relationships were discovered between the polymorphism in the *MTRR* A66G gene and the predisposition to colorectal cancer among Caucasians than East Asians [32]. Interestingly, Kim et al. found that increased risk of myelodysplastic syndrome and acute myelogenous leukemia rather than CML among the *MTRR* 66 AG genotype [24]. Few other studies have also examined the relationship between *MTRR* 66 AG polymorphism and cancer risk. These studies found a significant association between a decreased risk of acute lymphoblastic leukemia and an augmented risk for hepatocellular carcinoma [32,33,34,35,36]

Furthermore, inconsistent findings were published regarding the association between the *MTRR* 66 A > G (rs1801394) variant allele and the risk of breast cancer. While Hu et al. concluded no association [32]. Another study indicated that the *MTRR* polymorphisms were linked to postmenopausal women’s vulnerability to breast cancer [37]. However, these findings conflict with our report. Another study demonstrated that the association between *MTRR* and several types of cancer had dissimilar findings; the causes for these contradictory results are vaguely understood, but the discrepancies may be due to differences in the genetic background and/or in features related to the various populations investigated gene-environment interactions, plus small numbers of cases and controls [38]. This result accords with Yang et al. [39] but is inconsistent with studies conducted by Flores et al. [32,40]. However, our data fail to show a significant difference in gender and age for *MTRR* polymorphism among cases. Our study results suggested that *MTRR* A66G polymorphism was not showed a significant association with CML susceptibility in the Sudanese population.

## 5. Conclusions

In conclusion, our studies reported that the *MTRR* 66AG and GG are both associated with the decreased risk of developing CML in the Sudanese population. Given the relatively small size of our samples, further studies on larger cohorts with the determination of homocysteine level and folate situation are needed to confirm the impact of *MTRR* polymorphisms and its association with CML, particularly in the Sudanese population.

## Figures and Tables

**Figure 1 genes-13-01729-f001:**
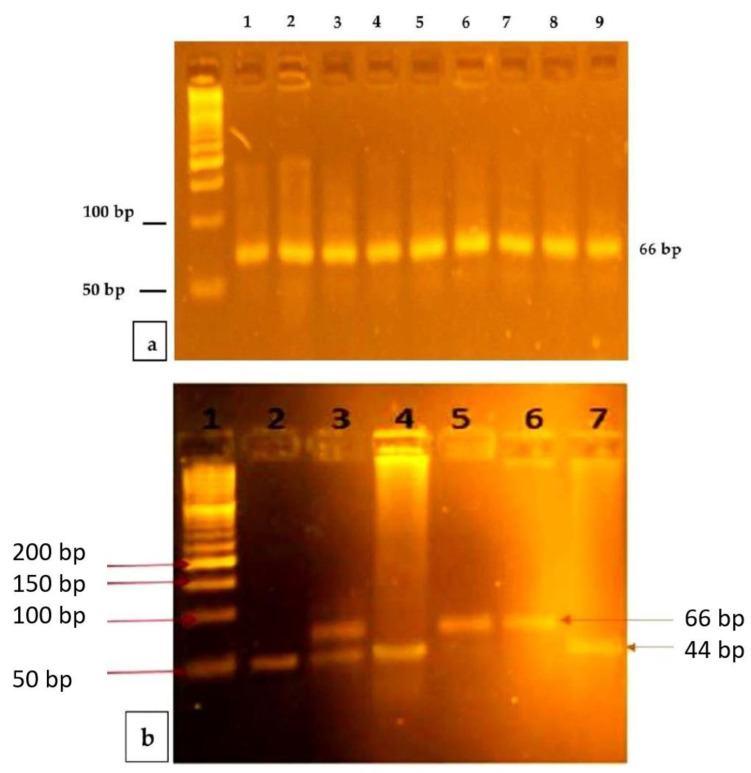
Representative PCR products and PCR-RFLP analysis of *MTRR* A66G. (**a**) PCR products of *MTRR* A66G: Lane 1 is a 50 base-pair ladder. Lanes 2–7 and 8 show the 66 bp PCR products. (**b**) PCR-RFLP analysis of *MTRR* A66G polymorphism using Nde1I restriction enzyme: Lane 1depicts the DNA molecular weight marker with a 50 base-pair ladder. Lanes 2, 4, and 7 show Homozygous wild-type (AA) one band at 44 bp (band 22 bp didnot appear due to its small molecular size). Lane 3 shows the Heterozygous (AG) two bands 66 and 44(also band 22 bp didnot appear due to its small molecular size). Lanes 5 and 6 show a Homozygous mutant (GG) one band 66 bp.

**Table 1 genes-13-01729-t001:** Comparison of demographic data according to *MTRR* genotype among patients.

Variables	*MTRR* AA	*MTRR* AG	*MTRR* GG	Total	*p*-Value
Age by years					
(Mean ± SD)	45.59 ± 12.3	43.64 ± 13.7	46.00 ± 10.7	45.06 ± 12.37	1.47
(Median)	46	44	47	45	1.4
Gender (n)%					
Males	44 (22.0%)	49 (24.5%)	39 (19.5%)	132 (66.0%)	0.178
Females	19 (9.5%)	20 (10.0%)	29 (14.5%)	68 (34.0%)	0.17

**Table 2 genes-13-01729-t002:** Comparing alleles and genotypes of *MTRR* A66G between CML patients and controls.

Genotype	CML N%	Controls N%	OR (95% CI)	*p*-Value
AA	6331.5	1313.0	Reference	
AG	6934.5	3939.0	0.365 (0.179–0.746)	0.006 *
GG	6834.0	4848.0	0.292 (0.145–0.590)	0.001 *
AG + GG	13,768.35	8787.0	0.146 (0.162–0.662)	0.002 *
A allele	19,548.75	6532.5	Reference	
G allele	20,551.25	13,567.5	0.506 (0.355–0.7292)	0.000 *

* = significant; OR = odd ratio and CI = confidence interval.

## Data Availability

The authors of this manuscript declare that data supporting the results of the current study are available within the article.

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
