# Peer review of "Impact of Methionine Synthase Reductase Polymorphisms in Chronic Myeloid Leukemia Patients"

_genes, 2022, doi:10.3390/genes13101729_

Round 1

Reviewer 1 Report

Abozer Y Elderdery and Collegaues presented the results of a project aiming to evaluate the impact of a MTRR polymorphism in the predisposition to CML. Despite the hot topic faced by the Authors, the project presents many limits and need to be strongly revised in order to support the reported conclusions.

1. Why did the Authors choose MTRR and no other gene involved in the same pathway?

2. Why did the Authors investigated only one polymorphism? Why A66G? MTRR presents many polymorphisms and mutations. The Authors should approach to their hypothesis querying more than one polymorphism. e.g. what about the haplotype? Is there any mutation known to segregate with the analyzed  polymorphism?

3. The age should be reported as median and range instead of mean and SD, because of its distribution.

4. Did the Authors considered any environmental factors? e.g. Radiations. This aspect and any additional rearrangement must be reported as characteristics of the cohort of patients, together with known comorbidities both for patients and for healthy controls

5. The results must be confirmed by a second test by molecular biology techniques 

6. Is the type of BCR-ABL1 transcript related with the presence of the polymorphism? The same for any additional rearrangement.

7. The Figures are not clear. In particular, Figure 1 should report also the 22bp fragment but it is not present and the fragments of sample 10 are longer than 50bp of the marker. The image of the fragments' migration on gel is confusing.

8. Line 155. "p=0.0000" What does it mean?

I strongly suggest the Authors to revised the project and to resubmit the manuscript with additional data and experiments.

Author Response

Response to Reviewer 1:

Thank you for your valuable comments and suggestions on our manuscript. Please find the below answers to your comments.

  1. Why did the Authors choose MTRR and no other gene involved in the same pathway?
  • Methionine synthase reductase (MTRR), is one of the major key regulatory enzyme involved in the folate metabolic pathway which is required for the reductive methylation of cobalamin and cofactor of methionine synthase (MS) in the remythalation of homocysteine to methionine. MTRR plays a vital role in maintaining the active state of MS, genetic variation within the MTRR gene may be associated with chronic myeloid leukemia. Therefore we have chosen MTRR gene than other genes.
  1. Why did the Authors investigated only one polymorphism? Why A66G? MTRR presents many polymorphisms and mutations. The Authors should approach to their hypothesis querying more than one polymorphism. e.g. what about the haplotype? Is there any mutation known to segregate with the analyzed polymorphism?
  • The MTRR gene is located on chromosome 5 at 5p15.2-p15.3 and the most common polymorphism in MTRR gene is the substitution of isoleucine with methionine at position 22 (A66G; rs1801394). It has been reported that the polymorphism MTRR A66G (rs1801394) in the MTRR gene is associated with decreased enzyme affinity for MS, which is closely associated with leukemia risk. However, MTRR polymorphism has been widely investigated for risk of leukemia. However, reported associations between this polymorphism and the risk of developing leukemia are inconsistent and remain controversial. To further clarify the association of MTRR A66G polymorphism with leukemia risk, we conducted a research study on A66G polymorphism and its association with leukemia risk in eligible Sudanese patients.
  1. The age should be reported as median and range instead of mean and SD, because of its distribution.
  • The data related to age reported as median and range instead of mean in the revised manuscript.
  1. Did the Authors considered any environmental factors? e.g. Radiations. This aspect and any additional rearrangement must be reported as characteristics of the cohort of patients, together with known comorbidities both for patients and for healthy controls.
  • We have not considered any environmental factors for the present study. The present study has been precisely focused on MTRR A66G polymorphism. The chosen patients were not suffering from certain diseases including chronic myelomonocytic leukemia or other myeloproliferative disorders, and were negative for Philadelphia chromosome (Ph). 
  1. The results must be confirmed by a second test by molecular biology techniques 
  • The mutation of MTRR A66G was determined by using polymerase chain reaction –restriction fragment length polymorphism (RFLP).
  1. Is the type of BCR-ABL1 transcript related with the presence of the polymorphism? The same for any additional rearrangement.
  • Chronic myelogenous leukemia (CML) is defined at molecular level by BCL-ABL1 fusion gene generated from a translocation between chromosome 9q34 and 22q11.2, forming Philadelphia chromosome [Ph]. BCR-ABL1 is the only genetic abnormality in 90% of CML cases in chronic phase. As disease progresses, clonal evolution with additional chromosomal changes (ACAs) emerges.
  1. The Figures are not clear. In particular, Figure 1 should report also the 22bp fragment but it is not present and the fragments of sample 10 are longer than 50bp of the marker. The image of the fragments' migration on gel is confusing.

 After run the digested MTRR PCR product by Nde1 restriction enzyme on gel electrophoresis, if the result shows single single band (66 bp) that is means GG homozygotes mutant, and when the result shows two bands of 44 bp, and 22 bp that is mean homozygotes wild type but, in this figure, the 22bp does not appear due to their small size, if the band 44bp appear we assume the band 22 bp it present. For the fragments of sample 10 are longer than 50 because sometimes the samples on the peripheral well have given un accurate results during the run on the gel electrophoresis the image was revised.

  1. Line 155. "p=0.0000" What does it mean?
  • P=0000 it means strongly significant because it is less than 0.05

Reviewer 2 Report

Considering certain major flaws in the study, i recommend a major revision of the manuscript.

In this study, Elderdery et al., investigated if MTRR (Methionine Synthase Reductase) A66G could represent significant risk factor for the development of Chronic Myeloid Leukemia in the Sudanese population.

With this aim, they assessed 200 CML patients and 100 age and gender matched healthy controls by PCR/RFLP technique. Firstly, the authors checked if the prevalence of these genotypes would correlate with some clinical characteristics of the CML patients, but no correlation was found.
Then, they compared MTTR genetic profile in CML cases vs control subjects, which determined a significant protective role of AG & GG genotypes individually as well as in the combined manner. The authors concluded that
MTRR A66G polymorphim act as an important genetic modifier for CML in the studied population.

Despite that, the authors found some significant results with MTRR A66G polymorphim genotypes; the following concerns need to be addressed.

Major:

1. The manuscript is a good research, average technically sound, but it needs a profound revision by a native speaker. There are many English mistakes on the text.

2. In highlights section, the authors have mentioned in section 3 that through this study, they will be able to develop suitable therapeutic approaches to treat CML. This statement is an overexxageration and uncalled for keeping in view that the authors have not performed any functional study for MTRR A66G polymorphim to ascertain its role in disease development. Moreover, in the absence of any drug-gene interaction study, to claim establishment of new therapeutic modalities on the basis of an epidemiological study is totally unacceptable.

3. Methodology section line 87; the current study being a prospective case-control study conducted between 01/2018-08/2018 for which the authors have recruited 200 CML patients. This represents a very high incidence of CML compared to rest of the world. Furthermore,  the present report is a regional and a single centre study. Therefore, the authors must include a reference to support this high CML incidence.

4. Methodology section: line 91-93; the authors have mentioned that one of the inclusion criteria for participants (cases and controls) was ph negativity. Kindly review this statement.

5. Methodology section: Figure 1: The authors claimed that PCR amplification of MTTR gene using specific primers produced 66 bp PCR product. However, the supplied gel pic does not complement the statement as the shown PCR amplicons does not seem to be 66bp. Kindly check.    

6. Result section: The authors claimed a lower frequency of AG & GG in CML cases compared to controls. Moreover, in line 154-155, the authors reported increased frequency of G allele in healthy controls compared to CML cases  (67.5% vs 51.25%). In view of the above two observations, the authors have contradictly reported a 2 times risk of CML development associated with “G” allele which is wrong. I advice authors to review that data and the statistical calculation and rectify the error.

7. The discussion section is very concise. It should be elaborated and the authors must discuss their results in the light of many recent studies studies on the same subject.

8. Did the authors try to carry out any sampling/experimental design before conducting the study? Determining the optimal sample size for their study could provide readers with an adequate number of participants to detect significant, robust results!

Minor:

Introduction:

a. The following nomenclature for SNPs: MTRR66A>G is unfamiliar. The authors need to add the rs number, this international accession number used by researchers and databases.

b. Symbols for genes need to be italicized all over the manuscript (including the title, abstract), whereas for proteins, the symbols are regular.

Methods:

a. 1964 Declaration of Helsinki for working on human subjects should be included.

b.  Please describe the characteristics of the controls and include exclusion and inclusion criteria in a point by point manner.

c. The authors should include a description about the Quality Control Assurance for genotyping.

Author Response

Response to Reviewer 2:

Thank you for your valuable comments and suggestions to develop our manuscript.

Major:

  1. The manuscript is a good research, average technically sound, but it needs a profound revision by a native speaker. There are many English mistakes on the text.
  • The manuscript now has been corrected by a professor of English.
  1. In highlights section, the authors have mentioned in section 3 that through this study, they will be able to develop suitable therapeutic approaches to treat CML.This statement is an overexxageration and uncalled for keeping in view that the authors have not performed any functional study for MTRR A66G polymorphim to ascertain its role in disease development. Moreover, in the absence of any drug-gene interaction study, to claim establishment of new therapeutic modalities on the basis of an epidemiological study is totally unacceptable.
  • The sentence has been revised in section 3 [Our findings will help to increase the understanding of MTRR A66G polymorphism and its association with leukemia risk in Sudanese population] in the revised manuscript.
  1. Methodology section line 87; the current study being a prospective case-control study conducted between 01/2018-08/2018 for which the authors have recruited 200 CML patients. This represents a very high incidence of CML compared to rest of the world. Furthermore,  the present report is a regional and a single centre study. Therefore, the authors must include a reference to support this high CML incidence.
  • Recent references has been included to support the CML incidence in revised manuscript.
  • Hairong He, Gonghao He, Taotao Wang, Jiangxia Cai, Yan Wang, Xiaowei Zheng, Yalin Dong, Jun Lu. Methylenetetrahydrofolate reductase gene polymorphisms contribute to acute myeloid leukemia and chronic myeloid leukemia susceptibilities: evidence from meta-analyses. Cancer Epidemiol.  2014 Oct;38(5):471-8
  • de Jonge R, et al. Polymorphisms in folate-related genes and risk of pediatric acute lymphoblastic leukemia. Blood. 2009.

  • Entisar M Tebien, Abozer Y. Elderdery, Jermy Mills, Hiba B Khalil. Detection of Genetic polymorphisms of Methylene tetrahydrofolate reductase among Sudanese patients with chronic myeloid leukemia. P J M H S. (13)4,2019.

  1. Methodology section: line 91-93; the authors have mentioned that one of the inclusion criteria for participants (cases and controls) was ph negativity. Kindly review this statement.
  • Chronic myelogenous leukemia (CML) can have other concurrent additional cytogenetic changes, especially during disease progression. Additional chromosomal changes (ACAs) in CML often occur in Philadelphia chromosome (Ph)-positive cells and are associated with disease acceleration and treatment resistance. The Ph-negative clone has genetic abnormality that not only induces t(9;22) but also other chromosomal changes. When t(9;22) emerges, the growth advantage of CML cells masks cells with other chromosomal changes. Therefore, in this study, we have chosen patients with  negative for Philadelphia chromose (Ph).
  1. Methodology section: Figure 1: The authors claimed that PCR amplification of MTTR gene using specific primers produced 66 bp PCR product. However, the supplied gel pic does not complement the statement as the shown PCR amplicons does not seem to be 66bp. Kindly check.   
  • In figure 1.1 section (a) the MTRR specific primers produced 66 bp amplicon after PCR when compared with ladder 50bp the bands are lays above 50 bp of the ladder fragments.

  1. Result section: The authors claimed a lower frequency of AG & GG in CML cases compared to controls. Moreover, in line 154-155, the authors reported increased frequency of G allele in healthy controls compared to CML cases  (67.5% vs 51.25%). In view of the above two observations, the authors have contradictly reported a 2 times risk of CML development associated with “G” allele which is wrong. I advice authors to review that data and the statistical calculation and rectify the error.
  • The data and the statistical calculation were reviewed.
  • The frequency of the 66GG genotype was significantly higher (OR = 0.292, 95% CI: 0.145- 0.590, P = 0.001) among controls (48.0%) when compared to the CML group (34.0%). Also, GG genotype reduces predisposition to CML. While the G allele was higher in the heathy group (67.5%) when compared with cases (51.25%), with statistically significant difference (P = 0.000), as well as G allele was reducing the CML risk (OR = 0.506, 95% CI: (0.355-0.7292), P = 0.000).  

  1. The discussion section is very concise. It should be elaborated and the authors must discuss their results in the light of many recent studies on the same subject.
  • The discussion section is developed and included recent studies in the revised manuscript
  1. Did the authors try to carry out any sampling/experimental design before conducting the study? Determining the optimal sample size for their study could provide readers with an adequate number of participants to detect significant, robust results!
  • We have conducted experiments from the period 2018/10 to 2018/08 and chosen the sample size availability of patients during the period.

Minor:

 Introduction:

  1. The following nomenclature for SNPs: MTRR66A>is unfamiliar. The authors need to add the rs number, this international accession number used by researchers and databases.
  • The text has been revised as MTRR 66A>G (rs1801394)
  1. Symbols for genes need to be italicized all over the manuscript (including the title, abstract), whereas for proteins, the symbols are regular.
  • The gene name has been italicized throughout the manuscript.

Methods:

  1. 1964 Declaration of Helsinki for working on human subjects should be included.
  • The above suggested declaration has been included in the revised manuscript.
  1. Please describe the characteristics of the controls and include exclusion and inclusion criteria in a point by point manner.
  • All participants were Sudanese
  • A control group containing healthy, age and sex matched individuals participated in this study.
  • The selected patients were not suffering from diseases including chronic myelomonocytic leukemia, myeloproliferative disorders and negative for Philadelphia chromosome.
  • Cytogenic, haematological and BCR-ABL transcript analysis were used to diagnose CML patients.
  • All controls who refused to participate in the study were excludes
  1. The authors should include a description about the Quality Control Assurance for genotyping.
  • The description of quality control for genotyping was included in the methodology section of the revised manuscript.

Reviewer 3 Report

Impact of Methionine synthase reductase polymorphisms in chronic myeloid leukemia patients Authors: Abozer Y Elderdery *, Enisar M, Tebien, Fawaz O Alenazy, Ahmed M Elkhalifa, Manar G Shalabi, Anass M, Abbas, Hassan H. Alhassan, Chand B Davuljigari *, Jeremy Mills

I would like to thank the Editors for the opportunity to review this publication.

In this article Abozer Y. Elderdery  et al.  investigated the role of polymorphic variant A66G of  MTT gene in the risk of CML.  The study  idea is general interesting – we need to  try to finding  risk or protective genetic variants of different traits. I have a few of comments:

1.       In the summary of the work, the authors write that: “Additionally, AG, GG genotypes and G allele were risk factors for CML (line 28) while in the discussion: “ individuals with GG or AG genotype showed they were protected from CML risk” (line 162). These are contradictory results and authors need to standardise the correct interpretation of the genotyping results, taking into account in particular the OR value indicating the risk. I hope it’s just an editorial error. In addition, gene symbols should be written in italics in the text.

2.      In my opinion, information on the gene, the chromosome position, as well as information on the localization of the polymorphic variant and its possible effects on gene expression, splicing, etc. should be included. Authors should also use a reference font for the polymorphic variant (rs number of variant) that is compatible with the HGVS.

3.      In the line 147 authors have to improve gene symbol (not MRR but MTRR).

                     I hope that my suggestions will be  helpful.

Author Response

Response to Reviewer 3:

Thank you for your valuable suggestions and comments on our manuscript. Please find the below answers to your comments.

. 1. In the summary of the work, the authors write that: “Additionally, AG, GG genotypes and G allele were risk factors for CML “ (line 28) while in the discussion: “ individuals with GG or AG genotype showed they were protected from CML risk” (line 162). These are contradictory results and authors need to standardize the correct interpretation of the genotyping results, taking into account in particular the OR value indicating the risk. I hope it’s just an editorial error. In addition, gene symbols should be written in italics in the text.

  • The statement now has been corrected in the revised manuscript and we have also corrected gene symbol as suggested.
  • The result was revised in the table 2.
  1. In my opinion, information on the gene, the chromosome position, as well as information on the localization of the polymorphic variant and its possible effects on gene expression, splicing, etc. should be included. Authors should also use a reference font for the polymorphic variant (rs number of variant) that is compatible with the HGVS.
  • We have included information on the gene, chromosome position and localization of the polymorphic variant in introduction section of the revised manuscript and corrected polymorphic variant information as suggested.
  1. In the line 147 authors have to improve gene symbol (not MRR but MTRR).
  • We have corrected the gene symbol accordingly

Round 2

Reviewer 1 Report

The Authors tried to reply to some of my questions, but several limits are still present.

1. The scientific reasons about the consideration of  MTRR and A66G must be clearly reported in the text.

2. The age must be reported as median and range. Mean, SD and median has no sense. Moreover, the data do not change (Hopefully!), but the authors reported two different p value: one about age expressed as mean and one about age expressed as median. What does it mean?

3. RFLP is the technique the Authors used to produce the primary data of the present study. As previously required, the data must be confirmed by a second molecular biology technique.

4. Probably my question about the BCR-ABL1 transcript type was not clear. I was wondering if the presence of BCR-ABL1 b2a2 and b3a2 is influenced by the presence of the studied variant.

5. If the Authors evaluated the 44bp band as proof of the presence of the variant, what about the Sample 3? It presents both the bands... Moreover the authors declared the difficulty in the presentation of the data and justify the wrong position of the band due to technical limit. Facing this elements, the confirmation of the data by a second molecular technique is mandatory.

Author Response

Answer for the Editor’s Comments

Manuscript number: ID 1807235

“Impact of Methionine synthase reductase polymorphisms in chronic myeloid leukemia patients

Thank you very much for your valuable suggestions. We have revised the manuscript based on the reviewers’ suggestions. The Authors tried to reply to some of my questions, but several limits are still present.

  1. The scientific reasons about the consideration of  MTRR and A66G must be clearly reported in the text.

We have included the following suitable reason about consideration of MTRR gene in the revised manuscript.

The MTRR gene is located on chromosome 5 at 5p15.2-p15.3 and the most common polymorphism in MTRR gene is the substitution of isoleucine with methionine at position 22 (A66G; rs1801394). It has been reported that the polymorphism MTRR A66G (rs1801394) in the MTRR gene is associated with decreased enzyme affinity for MS, which is closely associated with leukemia risk. However, MTRR polymorphism has been widely investigated for risk of leukemia. However, reported associations between this polymorphism and the risk of developing leukemia are inconsistent and remain controversial. To further clarify the association of MTRR A66G polymorphism with leukemia risk, we conducted a research study on A66G polymorphism and its association with leukemia risk in eligible Sudanese patients.

  1. The age must be reported as median and range. Mean, SD and median has no sense. Moreover, the data do not change (Hopefully!), but the authors reported two different p value: one about age expressed as mean and one about age expressed as median. What does it mean?
  • The data related to age reported as median and range instead of mean in the revised manuscript.

  1. RFLP is the technique the Authors used to produce the primary data of the present study. As previously required, the data must be confirmed by a second molecular biology technique.
  • Once again we have reported clear RFLP results by reexamining the data due to limitations ofour study.
  1. Probably my question about the BCR-ABL1 transcript type was not clear. I was wondering if the presence of BCR-ABL1 b2a2 and b3a2 is influenced by the presence of the studied variant.
  • The most common breakpoint cluster region gene-Abelson murine leukemia viral oncogene homolog 1 (BCR-ABL) transcripts in chronic myeloid leukemia (CML) are e13a2 (b2a2) and e14a2 (b3a2).Patients with CML express e13a2 or e14a2 mRNAs that result from a rearrangement of the major breakpoint cluster region (M-bcr) generating the 210-kDa (p210BCR-ABL) fusion proteins b2a2 or b3a2 respectively. The b3a2 variant is produced by the fusion of exon 14 of BCR gene with exon 2 of ABL1 gene while the b2a2 variant is the product of a fusion of BCR exon 13 and ABL1 exon 2. Several reports have suggested that the type of the chimeric mRNA (b2a2 or b3a2) type is associated with differences in the clinical and hematological characteristics of CML patients. However, this study particularly more focused on MTRR A66G gene and its association with CML risk in the Sudanese population. Therefore we will extend our future studies toinvestigate the influence of BCR-ABL1 mRNA transcripts (b2a2 and b3a2) by selected variant.
  •  

  1. If the Authors evaluated the 44bp band as proof of the presence of the variant, what about the Sample 3? It presents both the bands... Moreover the authors declared the difficulty in the presentation of the data and justify the wrong position of the band due to technical limit. Facing this elements, the confirmation of the data by a second molecular technique is mandatory
  • We included revised results and image in the revised manuscript. Further bands are very clear in results section. Please see the revised manuscript.
